# Body Mass Index (BMI) and Work Ability in Older Workers: Results from the Health and Employment after Fifty (HEAF) Prospective Cohort Study

**DOI:** 10.3390/ijerph17051647

**Published:** 2020-03-03

**Authors:** Catherine H Linaker, Stefania D’Angelo, Holly E Syddall, E Clare Harris, Cyrus Cooper, Karen Walker-Bone

**Affiliations:** 1MRC Versus Arthritis Centre for Musculoskeletal Health and Work, University of Southampton, Southampton General Hospital, Southampton SO16 6YD, UKech@mrc.soton.ac.uk (E.C.H.); 2Medical Research Council Lifecourse Epidemiology Unit, University of Southampton, Southampton General Hospital, Southampton SO16 6YD, UK

**Keywords:** body mass index, older worker, obesity, work ability, health-related job loss, sickness absence

## Abstract

This study explores associations between BMI and prolonged sickness absence; cutting down at work; and health-related job loss (HRJL) over two years of follow-up among workers aged ≥50 years. A cohort of 2299 men and 2425 women (aged 50–64 years) self-reported height and weight at baseline and provided information about work ability at 12 and 24 months for the Health and Employment after Fifty (HEAF) Study. Associations between BMI and work ability were assessed by logistic regression and HRJL by multiple-record Cox’s proportional hazards models, with adjustment for other risk factors. The prevalence of obesity/severe obesity was 22.6%/1.2% amongst men and 21.4%/2.6% amongst women, respectively. In men and women, obesity and severe obesity predicted having to cut down at work for health over two years. In women, severe obesity predicted prolonged sickness absence, and also HRJL even after adjustment for age, proximity to retirement, financial difficulties, and lifestyle factors (hazard ratio [HR] 2.93, 95% CI 1.38, 6.23), and additional adjustment for health conditions (HR 2.52, 95% CI 1.12, 5.67). Obesity, and particularly severe obesity, negatively impacts work ability amongst people aged 50–64 years, with greatest effects in women. Obesity can be expected to hinder attempts to encourage work to older ages.

## 1. Introduction

Obesity is a major and growing public health problem, with future global projections estimating that there will be more than one billion people affected by 2030 [1]. Obesity and overweight are linked to more deaths worldwide than underweight (except in parts of sub-Saharan Africa and Asia) because they are major risk factors for non-communicable diseases including: diabetes; cardiovascular diseases; musculoskeletal disorders, and common mental health conditions [2,3,4]. Although prevalence rates in children and adolescents are climbing, the highest prevalence is seen amongst men and women in the fifth, sixth, and seventh decades of life [5]. Contemporaneously, in Western countries, there have been dramatic changes in population demographics caused by increased longevity coupled with declining birth rates. This has led to reshaping of populations with increased proportions of economically inactive older people. For example, in Europe, it is projected that by 2050, about 30% of the population will be aged >65 years of age as compared with 20% in 2018 [6]. Governments have increasingly recognised that the resources set aside for pensions will be insufficient and have made policy changes to encourage people to stay in paid work to older ages. The Organisation for Economic Cooperation and Development OECD currently defines “older workers” as workers aged ≥50 years [7] and there is already evidence that these numbers are growing. For example, in the UK in 2019, more than 10 million people were “older workers” [8]. 

The impact of government policies to encourage individuals to postpone their retirement may be reduced if obesity limits work ability (defined as a worker’s capacity to sustain employment in his or her current job, given the demands of the job and his or her individual resources) [9], thereby increasing the likelihood of premature exit from the labour market. Previous research has indicated that obesity may impact on an individual’s work ability [10,11,12,13,14,15,16,17,18] which could, in turn, increase indirect societal costs (those resulting from reduced work productivity) [19,20,21]. According to at least one estimate, these may be greater than the direct health-care costs of obesity [22].

A recent British Government review concluded that evidence about the impact of obesity on employment is sparse, and that further research is urgently required [23]. We hypothesised that obesity might reduce work ability amongst older workers, thereby limiting their chances to work productively into their late 60s or beyond. We investigated this hypothesis by analysing longitudinal data from a population-based cohort study of health and work in later life, the Health and Employment After Fifty (HEAF) study. 

## 2. Materials and Methods 

The design and methods of the HEAF study have been reported previously [24]. Written informed consent was obtained from all participants. Ethical approval was provided by the National Health Service (NHS) Research Ethics Committee North West-Liverpool East (REC ref: 12/NW/0500).

### 2.1. Population Sampling

Briefly, over an 18-month period commencing in January 2013, a postal questionnaire was sent to 39,359 adults aged 50–64 years from 24 English general practices that contributed data to the Clinical Practice Research Datalink (CPRD). The CPRD is an electronic health record which collects de-identified coded patient data from a network of general practices across the UK. The data encompass 45 million patients, including 13 million currently registered patients. The questionnaire included questions about: marital status; level of education; proximity to retirement; levels of physical activity; alcohol consumption; smoking; home ownership; and self-perceived difficulty managing financially. 

Participants were asked about whether they were currently in paid employment and about their working conditions including: shift work; job satisfaction; coping with the mental and physical aspects of their job; and about the physical demands of their work (exposure in an average working day to: kneeling/squatting for >1 hr/day; climbing a ladder; digging/shovelling; lifting ≥10 kg by hand; hard physical working sufficient to cause sweating; and standing or walking for >3 hrs). The series of physical work demand questions were then summed (score 0–6).

### 2.2. Body Mass Index

Respondents were asked to self-report height (in cm or feet and inches) and weight (in kg or lbs.). Body mass index (BMI) was calculated as self-reported weight (kg) divided by the square of self-reported height (in metres), and classified in accordance with the international classification system of the World Health Organisation (WHO) [25]: underweight (<18.5 kg/m^2^); normal weight (18.5–24.9 kg/m^2^); overweight (25.0–29.9 kg/m^2^); obese (30–39.9 kg/m^2^); and obese class III (herein referred to as ‘severely obese’) (≥40 kg/ m^2^).

### 2.3. Prolonged Sickness Absence and Cutting Down at Work

Follow-up questionnaires were sent by post at 12 and 24 months after baseline to all participants who provided contact details and consented to follow-up. Each of these questionnaires collected information about: any period of sickness absence lasting more than 20 days over the past 12 months (‘prolonged sickness absence’), and whether they had needed to cut down, avoid, or change what they normally did at work because of a health problem (‘not at all’, ‘a little’, or ‘a lot’; analysed as ‘a lot’ vs. ‘not a lot’). 

### 2.4. Health-Related Job Loss (HRJL)

In both follow-up questionnaires, participants were asked if they had left a job within the preceding 12 months, the date that this occurred, and the reason for leaving. Health-related job loss was defined as a job loss attributed by the participant as having occurred “mainly” or “partly” for health reasons. 

### 2.5. Linked Health Data

Data linkage with the CPRD permitted access to objectively-recorded baseline information about a range of medical diagnoses (cardiovascular diseases, hypertension, stroke, diabetes, musculoskeletal disorders, common mental health problems, and severe mental health conditions) and drug prescriptions. In the current analyses, diagnoses of hypertension and common mental health conditions required substantiation with an appropriate drug prescription in the preceding 12 months.

### 2.6. Statistical Analysis

The total response rate to the baseline HEAF questionnaire was 8134 (20.7%). Of these, 7901 (97%) provided information on height and weight. After further exclusion of 88 participants with a BMI <18.5 kg/m^2^, 537 without CPRD linkage, and 587 who only returned the baseline questionnaire, there were 6689 eligible participants. Amongst these, 1715 never held a paid job at any of the baseline, first- or second-year follow-ups, and consequently the maximum analysis sample comprised 4974 participants.

Analysis on prolonged sickness absence and cutting down at work was restricted to 4178 and 4175 individuals respectively who provided information about these specific variables. Analysis on risk of health-related job loss (HRJL) over 2 years was restricted to the 4724 participants (2299 men) in work at some point between baseline and follow-ups and who provided sufficient information about dates of employment and work exits for inclusion in a multiple record survival dataset. 

Summary statistics were used to describe baseline characteristics according to BMI categories. Chi-squared tests were used to compare the differences between categories of BMI.

We first explored the association between BMI and prolonged sickness absence or needing to cut down, avoid, or change what they did at work a lot because of a health problem during the 2 years of follow-up. We used logistic regression with results expressed as odds ratios (ORs) with 95% confidence intervals (95% CIs). Finally, we used time to first event Cox’s proportional hazards models to analyse risk factors for HRJL at any time over 2 years of follow-up. Results were summarised by hazard ratios (HRs) and 95% CIs. 

For both methods, we used an iterative forward selection model building strategy to select covariates that should be adjusted for in multivariate models; this results in slightly different adjustment factors depending on outcome and sex. Analyses were conducted separately for men and women because men and women undertake different types of occupations: e.g., men are more likely to be employed in construction/manufacture, whilst women are more often employed in education, health, and social care, or the retail sectors. All statistical analyses were conducted using Stata (v 15.0) software (StataCorp LP, College Station, TX, USA).

## 3. Results

At baseline, the prevalence of obesity and severe obesity among men was 22.6% and 1.2% respectively, and was 21.4% and 2.6% among women respectively. Men and women with obesity or severe obesity were: less likely to own their own homes; more likely to report that they were struggling financially; more likely to do no leisure-time physical activity; less likely to drink any weekly alcohol and more likely to report that they were struggling to cope with the physical demands of their work (Table 1; Table 2). Men with obesity or severe obesity were more likely to be unmarried, while men with obesity were more likely to be ex or current smokers. Women with obesity or severe obesity were better educated and were more likely to be earning at least half of the household income, compared with those with normal weight. Likewise, the prevalence of hypertension, diabetes, musculoskeletal disorders, and common mental health conditions was increased amongst men and women with obesity and severe obesity. Additionally, women with obesity and severe obesity tended to be less satisfied with their job overall. 

### 3.1. Cutting Down at Work and Prolonged Sickness Absence

In total, across 2 years of follow-up, 206 (10.4%) men and 276 (12.6%) women experienced at least one episode of long-term sickness absence and 176 (8.9%) men and 182 (8.3%) women needed to cut down, avoid, or change what they usually did at work because of a health problem. Women with severe obesity reported increased odds of having had to cut down, avoid, or change what they did at work because of a health problem, even after adjustment for age, socio-economic, lifestyle factors, and either work factors or health (Table 3). No significant associations were reported among men after adjustment for the relevant factors. 

Amongst men, the odds of prolonged sickness absence were slightly increased amongst those with obesity. However, the association was attenuated by adjustment for health conditions. Women with obesity or severe obesity had greater odds of prolonged sickness absence compared with women of normal weight. Even after adjustment for socio-economic, lifestyle, and either work factors or health, these associations remained elevated (ORs (95%CI) =1.7 (1.2, 2.3) and =2.2 (1.1, 4.4) for women with obesity and severe obesity respectively). 

### 3.2. Health-Related Job Loss (HRJL)

A total of 101 (4.4%) men and 152 (6.3%) women experienced a health-related job loss over the two-year follow-up. Amongst women, risk of HRJL was somewhat elevated across all categories of BMI ≥ 25 kg/m^2^ in comparison with women of normal weight (Table 4, model 1); however, adjustment for age, sociodemographic characteristics, and lifestyle risk factors showed that this increased risk was focused amongst women with severe obesity (HR = 2.93; 95%CI 1.38 to 6.23). We found no convincing evidence for an association between BMI and risk of HRJL among men. 

In univariate analysis (data not shown), baseline obesity was associated with job dissatisfaction (only among men) and physical activity score at work (both sexes) and therefore, we further adjusted the analyses for these factors (model 3). Once again, compared with women of normal weight, women with severe obesity were almost three times as likely to experience HRJL (HR 2.98, 95%CI 1.40–6.33).

Amongst women, musculoskeletal disorders and common mental health conditions were independent risk factors for HRJL (data not shown). Therefore, in model 4, we adjusted for the effects of these conditions and found that women with severe obesity remained at increased risk of HRJL compared to women of normal weight (HR = 2.52; 95%CI 1.12 to 5.67). This increased risk for women with severe obesity persisted when we mutually adjusted for both work and health conditions (model 5).

## 4. Discussion

In this longitudinal study, obesity, and particularly severe obesity, predicted difficulties coping at work amongst older workers. Amongst men and women, severe obesity predicted needing to cut down, avoid, or change what they did at work over two years of follow-up. Compared with those of normal weight, a higher risk of prolonged sickness absence during two years of follow-up was seen amongst women with obesity or severe obesity. Women with a BMI ≥25 kg/m^2^ were at greater risk of HRJL during two-year follow-up when compared with those of normal weight; however, only women with severe obesity had an elevated risk of HRJL (more than 2.5-fold increased) after adjustment for other risk factors. These findings add to a growing body of literature which suggests that obesity may impair work ability, adding to it by studying several different work outcomes longitudinally and focusing on workers aged ≥50 years at a time when societies are looking to encourage people to stay in work to older ages.

Obesity and work ability have been studied previously. In a cross-sectional study including 10,400 Danish workers, mean age 43.5 years (range 18–59 years), work ability defined as “coping with physical demands” was found to be impaired in association with obesity with estimated risks increasing with increasing BMI [26], but to our knowledge, our finding that severe obesity predicted needing to cut down, avoid, or change what individuals can manage at work, as well as health-related job loss, over the next two years is novel. Interestingly, Gates and colleagues [16] evaluated presenteeism amongst >300 workers in the manufacturing sector using the Work Limitations Questionnaire and found that severe obesity was associated with health-related work limitations, particularly regarding time needed to perform tasks and ability to cope with the physical demands of work. Moreover, in a systematic review, Goettler and colleagues [19] reported the high rate of indirect costs associated with overweight and obesity attributable to the burden of lost productivity as well as absenteeism. 

Sickness absence associated with overweight and obesity has been more widely studied [12,13,14,15,18,26,27,28], and it has been estimated that individuals with obesity take an extra four days of sick leave annually compared with employees of a normal weight [14]. Such is the extent of this literature that there have been three systematic reviews [29,30,31], each of which suggest that overweight and obesity are associated with increased sickness absence. Whilst the data conflict somewhat regarding the relationship with short-term sickness absence, studies consistently demonstrate that obesity predicts sickness absence of >7 days [31]. Although extensively studied across the working-age range, we found few data specific to older workers with which to compare the current results. It is important to interpret our findings in light of the fact that, to be eligible for this analysis, people had to be in paid work at baseline or the first year of follow-up in order for their future sickness absence to be evaluated. This suggests that this cohort have, to some extent, exhibited a “healthy worker” effect at least until the age of 50 years and that, even then, it is possible to observe an impact of obesity on future absenteeism. Rates of return to any type of work from sick leave decline progressively with increasing duration of sickness absence, and long-term sick leave is an important predictor of future unemployment [32]. Hence, we chose to study prolonged sick leave (>20 days) as a marker of risk of future work disability. It is, therefore, an important finding that, amongst women, obesity was an important predictor of prolonged absenteeism. 

Numerous studies have investigated the relationship between disability pensioning and overweight and obesity [33,34]. A meta-analysis of 28 studies reported that the relative risk of disability pensioning was 1.53 for individuals with obesity and 1.16 for those overweight [35]. Similar results were reported more recently in a meta-analysis in which the outcome was “disability retirement” and which demonstrated that obesity was also associated with disability retirement caused by musculoskeletal disorders, cardiovascular diseases, and mental disorders [36]. Unfortunately, fundamental differences in the structuring of welfare systems make comparison unfeasible between countries regarding disability pensioning and disability retirement. For example, in some countries, all paid employees have insurance offered by the employer that provides a disability pension when and if required. In the UK, only a minority of workers has access to privately paid health insurance and therefore, people who stop working because of their health before the state pension age frequently rely on either private finance or state welfare (“benefit”) payments. For this reason, we decided to study “health-related job loss” in order to identify people who attribute their inability to work at least partly to their health, irrespective of their financial circumstances or insurance provision. Our results indicate that amongst older female workers, severe obesity more than doubles the risk of health-related job loss. To our knowledge, the only similar data are from a study by Houston and colleagues [37] who showed that obesity at age 45–55 years predicted retirement before the age of 65 years due to a health reason in men and women from white and African American backgrounds. Interestingly however, their study demonstrated the highest risk was amongst white men (HR: 4.29 (95%CI 2.28–8.07)), with similar risk estimates to those attained in the current study, although, in their study, participants were not necessarily in paid work at the time BMI was calculated. In another study of older workers, severe obesity predicted exit from paid work through unemployment (OR 1.92, 95%CI 1.21–3.07) and retirement (OR 1.40, 95%CI 1.05–1.84) but not disability [38]. Thus, despite some variation in the outcome measure, the message is consistent: severe obesity increases the risk of exit from paid employment amongst older workers.

One could postulate two possible mechanisms by which obesity influences workability: firstly, the shape/size and physical capacity of the individual affects their ability to undertake the physical demands of work; or secondly, through ill-health and comorbidity resulting from obesity. In our analyses, we adjusted both for physical work demands and for those health conditions (musculoskeletal disorders and common mental health conditions) which were found to be independently associated with health-related job loss. It is interesting that a significant effect of obesity remained, even after adjustment for these variables. This suggests either additional health impacts of the obesity which are not yet diagnosed or manifest, or indeed, that other factors relating to obesity exist that impact workability amongst older female workers. Stigma, discrimination, impaired body image [39], or poorer coping skills, might all play a role [40]. Certainly, if an older worker is struggling to cope at work, they may opt to retire in preference to finding alternative employment. This may be particularly pertinent in the case of obesity, which is likely to adversely affect an individual’s body image and their perceived likelihood of being successful at competitive interview, given the discrimination which is ubiquitous in the workplace.

Our findings need to be considered alongside a number of limitations. Firstly, the prevalence of overweight, obesity, and severe obesity was based upon self-reported height and weight data. It is recognised that self-reported sampling may lead to underestimation of the true BMI [41]. However, as the >8000 HEAF participants were deliberately sampled all across England, we were logistically unable to objectively measure these parameters. The prevalence of overweight and obesity may have been underestimated if those who had a high BMI were less likely to return a questionnaire. To mitigate this risk, the questions about height and weight were in the “lifestyle” section of the questionnaires and were not emphasised to the participants within a range of questions which were specifically focused around health and work. Therefore, we believe that any errors in the assessment of BMI would be expected to be non-differential, and therefore to have obscured any true associations with employment outcomes. It is important to note however, that we found a relatively low prevalence of severe obesity amongst the male respondents, which may have limited our statistical power to find associations in men that were seen in women.

Our longitudinal study has the advantage that it employed a rigorous population-based sampling strategy with a broad geographical representation. Almost everyone in England registers for primary care services, making GP lists an excellent sampling frame for population-based epidemiological studies. However, despite the large achieved sample size, it must be borne in mind that only around 20.7% of the adults aged 50–64 years who were registered with each of the sampled practices consented to enter the. Responders were somewhat older, more affluent, and more often female than non-responders, although the sample’s demographic profile approximated reasonably to national statistics for the age band under study and included the full range of socio-economic deprivation [24]. The analyses presented here rely upon internal comparisons drawn amongst participants within the HEAF cohort and importantly for internal comparisons, the retention rate to the HEAF study has been excellent (84% and 83% at the first and second year follow-ups respectively), and therefore there is a low risk of attrition bias having led to biased results.

Whilst the HEAF study benefits greatly from the access to objective health diagnoses for the majority of participants via the CPRD, it is important to be aware that CPRD records may contain numerous entries for each individual patient. In some cases, the CPRD record for HEAF participants went back for more than 20 years. One accepted limitation of the CPRD is that there is heterogeneity in terms of diagnostic coding quality, since many healthcare professionals allocate codes based upon their individual assessment of a particular consultation. The governance of CPRD does include a regular assessment of fidelity of coding for some objective conditions but it is well-recognised that it is more difficult to substantiate diagnoses with less objective criteria, e.g., musculoskeletal pain and osteoarthritis [42]. For this reason, we focused our analyses on unambiguous diagnostic entities with good validity e.g., diabetes mellitus and cardiovascular disease [43]. For commonly-occurring mental health conditions which may be mild and fluctuating in nature, the analyses were deliberately restricted to consider only those diagnosed and/or treated within the past 12 months, in order to make them as relevant as possible to the current self-reported health data.

Finally, it is important to consider our findings alongside an awareness that obesity is a complex and multi-factorial condition. By definition therefore, it is simplistic to consider single outcomes (in this case work ability) in association with obesity as if they were isolated exposures and outcomes without a complex causal pathway. In the current study, we present longitudinal observational data collected from a population sample of contemporary workers in order to shed new insights for generation of hypotheses for future testing. As expected, [44], we found that obesity was strongly associated with other markers of lower socio-economic position (SEP): educational attainment; struggling financially; and lack of home ownership. Marmot and others have highlighted the complexities underpinning the social determinants of health [45]. However, our aim was solely to shed light on the relationship between obesity and ability to work to older ages at a time when policy makers have changed policy in the assumption that everybody can be expected to be able to continue in paid work into their 7 th decade of life. People from a poorer SEP are more likely to have physically-demanding occupations and less likely to have robust private pension arrangements. The implication is that obesity, yet another consequence of low SEP, adds to the risk of premature loss of personal economic productivity, resulting in more individuals dependent upon state welfare (indirect costs), experiencing poorer health (consequently higher direct health costs), and furthermore in a precarious position in respect of their home circumstances. Governments that have increased the age of eligibility for state pensions (such as the UK and Denmark), have made such wholesale changes without considering the nature and type of work that individuals are expected to carry out into their seventh decade of life. Our findings suggest that any fiscal savings achieved by raising state pension age may well be negated by increased health and welfare costs. 

## 5. Conclusions

Overall, our study supports an association of obesity, and particularly severe obesity, with adverse employment outcomes, particularly in older female workers. The high and growing population burden of obesity can be expected to hinder attempts to encourage work to older ages. There is a growing incentive for employers to introduce measures designed to help employees maintain a healthy weight.

## Figures and Tables

**Table 1 ijerph-17-01647-t001:** Baseline characteristics of men, by categories of body mass index (BMI).

Characteristics	BMI (kg/m^2^)				
	18.5–24.9 (Normal)	25–29.9 (Overweight)	30–39.9 (Obese)	≥40 (Severely Obese)	*p* ^2^
	N = 626 (27.2%)	N = 1126 (49.0%)	N = 519 (22.6%)	N = 28 (1.2%)	
*Socio-demographic variables*					
Age (years), mean (SD)	57.6 (4.2)	57.8 (4.2)	58.0 (4.2)	57.7 (4.3)	0.58
Marital status—Not married	151 (24.2)	231 (20.5)	135 (26.1)	12 (44.4)	0.003
Proximity to retirement					
Less than a year	33 (5.6)	59 (5.5)	28 (5.8)	1 (3.9)	0.88
1 to <5 years	144 (24.2)	278 (25.8)	114 (23.8)	4 (15.4)	
5–10 years	199 (33.5)	352 (32.7)	172 (35.8)	12 (46.2)	
10+ years	219 (36.8)	389 (36.1)	166 (35.6)	9 (34.6)	
Level of education					
No qualification/School	168 (26.8)	340 (30.2)	165 (31.8)	7 (25.0)	0.12
Vocational training certificate	208 (33.2)	362 (32.2)	189 (36.4)	10 (35.7)	
University degree/higher	250 (39.9)	424 (37.7)	165 (31.8)	11 (39.3)	
Proportion of family income earned by you					
None	16 (2.6)	44 (4.0)	17 (3.4)	-	0.72
Less than a quarter	32 (5.2)	44 (4.0)	22 (4.3)	2 (7.4)	
Between a quarter and a half	89 (14.5)	159 (14.5)	74 (14.6)	2 (7.4)	
Half or more	477 (77.7)	853 (77.6)	394 (77.7)	23 (85.2)	
Social class					
Higher managerial	275 (44.6)	474 (42.9)	205 (39.9)	10 (35.7)	0.32
Intermediate occupation	135 (21.9)	232 (21.0)	100 (19.5)	6 (21.4)	
Routine occupation	206 (33.4)	400 (36.2)	209 (40.7)	12 (42.9)	
Home ownership—Rented/rent free	71 (11.5)	113 (10.2)	75 (14.6)	8 (29.6)	0.002
Managing financially—finding it difficult	41 (6.7)	69 (6.2)	50 (9.8)	8 (29.6)	<0.001
*Lifestyle factors*					
No physical activity	78 (13.3)	155 (15.0)	96 (20.1)	8 (32.0)	0.002
Alcohol consumption					
Low/no drinker (≤1 unit /wk)	71 (12.0)	121 (11.3)	92 (18.9)	7 (28.0)	<0.001
Moderate (2–14 units/wk)	331 (56.0)	560 (52.1)	230 (47.1)	14 (56.0)	
Heavy (15+ units/wk)	189 (32.0)	394 (36.7)	166 (34.0)	4 (16.0)	
Ex/current smoker	275 (44.1)	540 (48.1)	280 (54.3)	12 (42.9)	0.007
*Health from CPRD*					
Diabetes	32 (5.1)	104 (9.2)	90 (17.3)	6 (21.4)	<0.001
Musculoskeletal diagnoses	106 (16.9)	281 (25.0)	163 (31.4)	8 (28.6)	<0.001
Stroke/CVA or TIA—ever before	9 (1.4)	16 (1.4)	10 (1.9)	-	0.78
Common mental health and treatment in past 12 m	57 (9.1)	102 (9.1)	64 (12.3)	7 (25.0)	0.01
Hypertension and treatment in past 12 m	80 (12.8)	264 (23.5)	196 (37.8)	17 (60.7)	<0.001
Cardiac conditions—ever before	43 (6.9)	89 (7.9)	57 (11.0)	5 (17.9)	0.02
*Work factors* ^1^					
Overall job dissatisfaction	47 (7.7)	71 (6.6)	43 (8.6)	5 (18.5)	0.08
Physical work activities score (0–6), median (IQR)	1.0 (0, 3.0)	1.0 (0, 3.0)	1.0 (0, 3.0)	1.0 (0, 3.0)	0.84

^1^ For descriptive purposes in this table only, we describe work factors on the basis of baseline information for 2228 (97%) men and also for 71 men who were not in paid employment at baseline but were in work at the first annual follow-up. ^2^
*p* values from Chi-squared test.

**Table 2 ijerph-17-01647-t002:** Baseline characteristics of women, by categories of body mass index (BMI).

Characteristics	BMI (kg/m^2^)				
	18.5–24.9 (Normal)	25–29.9 (Overweight)	30–39.9 (Obese)	≥40 (Severely Obese)	*p* ^2^
	N = 1055 (43.5%)	N = 787 (32.5%)	N = 520 (21.4%)	N = 63 (2.6%)	
*Socio-demographic variables*					
Age (years), mean (SD)	57.1 (4.0)	57.4 (3.9)	57.0 (3.8)	57.3 (3.8)	0.19
Marital status—Not married	341 (32.7)	247 (31.7)	196 (38.1)	23 (37.1)	0.08
Proximity to retirement					
Less than a year	56 (5.8)	45 (6.1)	29 (6.0)	5 (8.5)	0.32
1 to <5 years	231 (23.9)	162 (22.1)	93 (19.3)	8 (13.6)	
5–10 years	356 (36.9)	275 (37.5)	187 (38.7)	30 (50.9)	
10+ years	323 (33.4)	251 (34.2)	174 (36.0)	16 (27.1)	
Level of education					
No qualification/School	344 (32.6)	302 (38.4)	198 (38.1)	16 (25.4)	0.02
Vocational training certificate	320 (30.3)	238 (30.2)	156 (30.0)	26 (41.3)	
University degree/higher	391 (37.1)	247 (31.4)	166 (31.9)	21 (33.3)	
Proportion of family income earned by you					
None	48 (4.7)	37 (4.8)	25 (5.0)	2 (3.3)	0.03
Less than a quarter	189 (18.6)	102 (13.3)	64 (12.7)	8 (13.1)	
Between a quarter and a half	252 (24.8)	205 (26.7)	118 (23.4)	12 (19.7)	
Half or more	527 (51.9)	425 (55.3)	298 (59.0)	39 (63.9)	
Social class					
Higher managerial	451 (43.2)	313 (40.2)	216 (41.7)	29 (46.0)	0.60
Intermediate occupation	351 (33.6)	254 (32.6)	171 (33.0)	18 (28.6)	
Routine occupation	243 (23.3)	212 (27.2)	131 (25.3)	16 (25.4)	
Home ownership—Rented/rent free	99 (9.6)	104 (13.4)	73 (14.4)	7 (11.5)	0.02
Managing financially—finding it difficult	81 (7.9)	71 (9.1)	58 (11.4)	9 (14.8)	0.06
*Lifestyle factors*					
No physical activity	126 (13.1)	123 (18.1)	120 (26.2)	18 (32.7)	<0.001
Alcohol consumption					
Low/no drinker (≤1 unit /wk)	220 (22.3)	192 (27.4)	181 (39.8)	25 (52.1)	<0.001
Moderate (2–14 units/wk)	678 (68.7)	460 (65.6)	245 (53.9)	20 (41.7)	
Heavy (15+ units/wk)	89 (9.0)	49 (7.0)	29 (6.4)	3 (6.3)	
Ex/current smoker	412 (39.2)	352 (45.0)	236 (45.5)	24 (38.1)	0.03
*Health from CPRD*					
Diabetes	25 (2.4)	44 (5.6)	65 (12.5)	13 (20.6)	<0.001
Musculoskeletal diagnoses	226 (21.4)	232 (29.5)	165 (31.7)	26 (41.3)	<0.001
Stroke/CVA or TIA—ever before	3 (0.3)	7 (0.9)	3 (0.6)	1 (1.6)	0.26
Common mental health and treatment in past 12 m	196 (18.6)	145 (18.4)	138 (26.5)	19 (30.2)	<0.001
Hypertension and treatment in past 12 m	106 (10.1)	133 (16.9)	124 (23.9)	26 (41.3)	<0.001
Cardiac conditions—ever before	19 (1.8)	22 (2.8)	15 (2.9)	3 (4.8)	0.25
*Work factors* ^1^					
Overall job dissatisfaction	55 (5.4)	39 (5.1)	42 (8.2)	6 (9.8)	0.05
Physical work activities score (0–6), median (IQR)	0 (0,1.0)	0 (0,1.0)	0 (0,1.0)	0 (0,2.0)	0.94

^1^ For descriptive purposes in this table only, we describe work factors on the basis of baseline information for 2347 (97%) women and for 78 women who were not in paid employment at baseline but were in work at the first annual follow-up. ^2^
*p* values from Chi-squared test.

**Table 3 ijerph-17-01647-t003:** Odds ratio (OR) and 95%CI for work outcomes within 2 years of follow-up by categories of body mass index (BMI) for men and women. (Bold type denotes statistical significance at the *p* < 0.05 level).

	OR (95%CI)									
	MEN					WOMEN				
	N (%) cases	Model 1	Model 2	Model 3	Model 4	N (%) cases	Model 1	Model 2	Model 3	Model 4
*Prolonged sickness absence* (M = 1983; W = 2195)										
18.5–24.9 (Normal)	42 (4.8)	Ref				97 (6.3)	Ref			
25–29.9 (Overweight)	107 (7.4)	**1.5 (1.0, 2.1)**	1.4 (0.9, 2.0)	1.4 (0.9, 2.1)	1.4 (0.9, 2.0)	78 (6.4)	1.1 (0.8, 1.5)	1.0 (0.8, 1.5)	1.0 (0.7, 1.5)	1.0 (0.7, 1.4)
30–39.9 (Obese)	54 (8.0)	**1.7 (1.1, 2.6)**	**1.6 (1.0, 2.4)**	**1.6 (1.0, 2.5)**	1.5 (0.9, 2.3)	87 (11.0)	**2.0 (1.4, 2.7)**	**2.0 (1.4, 2.7)**	**2.0 (1.4, 2.7)**	**1.7 (1.2, 2.3)**
≥40 (**Severely** Obese)	3 (7.0)	1.9 (0.5, 6.6)	1.7 (0.5, 6.1)	1.9 (0.5, 6.7)	1.3 (0.4, 4.9)	14 (13.7)	**2.9 (1.5, 5.5)**	**2.6 (1.3, 5.1)**	**2.6 (1.3, 5.1)**	**2.2 (1.1, 4.4)**
*Had to cut down work for health reasons* (M = 1981; W = 2194)										
18.5–24.9 (Normal)	37 (4.2)	Ref				69 (4.5)	Ref			
25–29.9 (Overweight)	77 (5.3)	1.2 (0.8, 1.8)	1.1 (0.7, 1.7)	1.2 (0.8, 2.0)	1.1 (0.7, 1.7)	57 (4.7)	1.1 (0.8, 1.6)	1.2 (0.8, 1.7)	1.2 (0.8, 1.7)	1.1 (0.8, 1.7)
30–39.9 (Obese)	57 (8.4)	**2.0 (1.3, 3.1)**	**1.7 (1.1, 2.8)**	**1.9 (1.1, 3.1)**	1.5 (0.9, 2.4)	44 (5.6)	1.3 (0.9, 1.9)	1.3 (0.9, 2.0)	1.4 (0.9, 2.2)	1.1 (0.7, 1.7)
≥40 (**Severely** Obese)	5 (11.6)	**3.9 (1.4, 11.1)**	2.0 (0.5, 7.6)	1.9 (0.5, 7.5)	1.3 (0.3, 5.1)	12 (11.8)	**3.4 (1.7, 6.7)**	**3.3 (1.6, 6.7)**	**3.4 (1.6, 7.2)**	**2.6 (1.2, 5.5)**

Prolonged sickness absence: *Model 1*: adjusted for age, *Model 2*: adjusted for age + household income + social class, *Model 3*: as Model 2 + physical work activity score, *Model 4*: as Model 2 + common mental health problems, musculoskeletal disorders; Having to cut down work a lot: *Model 1*: adjusted for age, *Model 2*: adjusted for age + household income + social class + finances + no physical activity + smoking status (men); age + finances + having no friends, *Model 3*: as Model 2 + job satisfaction + physical work activity score (men); as Model 2 + physical work activity score (women), *Model 4*: as Model 2 + common mental health problems + regional pain (men); as Model 2 + common mental health problems + regional pain + hypertension (women).

**Table 4 ijerph-17-01647-t004:** Hazard ratio (HR) and 95%CI for health-related job loss (HRJL) by categories of body mass index (BMI), for men and women. (Bold type denotes statistical significance at the *p* < 0.05 level).

	HR (95%CI)				
	Model 1	Model 2	Model 3	Model 4	Model 5
*Men*					
18.5–24.9 (Normal)	Ref	Ref	Ref	Ref	Ref
25–29.9 (Overweight)	1.07 (0.67, 1.73)	1.26 (0.75, 2.11)	1.21 (0.72, 2.04)	1.22 (0.72, 2.05)	1.17 (0.69, 1.98)
30–39.9 (Obese)	1.06 (0.61, 1.86)	1.20 (0.66, 2.18)	1.18 (0.65, 2.16)	1.04 (0.57, 1.93)	1.01 (0.55, 1.87)
≥40 (**Severely** Obese)	1.82 (0.43, 7.69)	1.53 (0.35, 6.75)	1.57 (0.36, 6.92)	0.83 (0.18, 3.84)	0.99 (0.22, 4.49)
*Women*					
18.5–24.9 (Normal)	Ref	Ref	Ref	Ref	Ref
25–29.9 (Overweight)	1.42 (0.98, 2.07)	**1.55 (1.04, 2.32)**	**1.56 (1.04, 2.32)**	1.26 (0.82, 1.95)	1.26 (0.82, 1.95)
30–39.9 (Obese)	1.34 (0.87, 2.05)	1.29 (0.81, 2.07)	1.31 (0.82, 2.10)	1.18 (0.72, 1.93)	1.19 (0.73, 1.94)
≥40 (**Severely** Obese)	**2.52 (1.20, 5.30)**	**2.93 (1.38, 6.23)**	**2.98 (1.40, 6.33)**	**2.52 (1.12, 5.67)**	**2.56 (1.14, 5.77)**

*Model 1*: adjusted for age, *Model 2*: adjusted for age, proximity to retirement, financial difficulties, home ownership, and physical activity for men; adjusted for age, proximity to retirement, financial difficulties, educational level, and smoking for women, *Model 3*: as Model 2 + job dissatisfaction, physical work activity score among men; as Model 2 + physical work activity score among women, *Model 4*: as Model 2 + hypertension, cardiac conditions, common mental health problems among men; as Model 2 + musculoskeletal disorders, common mental health problems among women, *Model 5*: as Model 3 + CPRD health conditions.

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
