# Peer review of "Body Mass Index (BMI) and Work Ability in Older Workers: Results from the Health and Employment after Fifty (HEAF) Prospective Cohort Study"

_ijerph, 2020, doi:10.3390/ijerph17051647_

Round 1

Reviewer 1 Report

Thank you for the authors' responses and revision. I do not have further comments. 

Author Response

We are grateful to Reviewer 1 for their support

Reviewer 2 Report

The modified version is fine.

Author Response

Thank you to Reviewer 2.

Reviewer 3 Report

The authors have done a good job and substantially improved the manuscript. I only found an error at the end of line 141; according to the results of table 1, the text must say not married, or table 1 (marital status) say married.

Author Response

We are extremely grateful to reviewer 3 for his thorough review on both occasions. He/she is quite correct, there is a typographic error in the text (not the table) and this should read UNmarried.

This has been corrected in the first paragraph of Results as follows:

At baseline, the prevalence of obesity and severe obesity among men was 22.6% and 1.2% respectively, and was 21.4% and 2.6% among women respectively. Men and women with obesity or severe obesity were: less likely to own their own homes; more likely to report that they were struggling financially; more likely to do no leisure-time physical activity; less likely to drink any weekly alcohol and more likely to report that they were struggling to cope with the physical demands of their work (Tables 1 and 2). Men with obesity or severe obesity were more likely to be unmarried, while men with obesity were more likely to be ex or current smokers. Women with obesity or severe obesity were better educated and were more likely to be earning at least half of the household income, compared with those with normal weight. Likewise, the prevalence of hypertension, diabetes, musculoskeletal disorders and common mental health conditions was increased amongst men and women with obesity and severe obesity. Additionally, women with obesity and severe obesity tended to be less satisfied with their job overall.

This manuscript is a resubmission of an earlier submission. The following is a list of the peer review reports and author responses from that submission.

Round 1

Reviewer 1 Report

This prospective cohort study investigated the associations between body mass index (BMI) and workability. Over 4000 employed men and women aged 50-64 years were included in the data analysis. BMI was self-reported at baseline and participants were followed up by questionnaires for outcomes of interest including prolonged sickness absence, cutting down at work, as well as health-related job loss. The study found that obesity reduces workability after multiple adjustment at older ages especially in women. I found the study overall well designed and properly conducted. I only have a few minor comments/questions.

Did the baseline questionnaire assess whether the participants had prolonged sickness absence or cutting down in the past (i.e., at baseline)? Would you consider briefly explaining why the data analysis was conducted separately for men and women? This information might be interesting to readers. The sample size for severe obesity is relatively small especially for men.

Author Response

Reviewer 1

Authors should change the way they approach the problem, including several variables in structural models, and test the existing models and theories. The information here provided is very simple and not useful.

As advised by the Editor, we have not made any changes to our primary analyses.

The problem of obesity is multifactorial, and its consequences equally multiple and correlated. I think that conducting studies on a single cause or consequence of obesity is currently completely irresponsible and biased since the variables overlap and hide between them, and more if the data were obtained by self-report. These types of studies should be carried out including many variables in multifactorial analysis and structural models, thereby seeing the influence and strength of each variable in the problem.
As observed in the results, the authors have other data that they should include in multifactorial analyzes and test the published models, reinforcing the published associations and/or providing new equations.
The association problems of only two variables are weak and subject to bias. Also, from a scientific point of view, partial publications, are biases and dishonest, creating many false positives and dangerous scientific myths.

See responses to Editor Comments above.

Reviewer 2 Report

Catherine et al studied the relationship between BMI and workability at older ages. Data from 2299 men and 2425 women (aged 50-64 years) self-reported height and weight at baseline and provided information about workability at 12 and 24 months for the Health and Employment after Fifty (HEAF) Study. They found that obesity, and particularly severe obesity, negatively impacts workability at older ages with greatest effects in female workers. This cohort study is an interesting study. There are several issues the authors need to address.

The data is collected in 2013. Could the author find some recent data. In table 1 and 2, how to calculate the P value, should be announced in the manuscript. In table 3 and 4, some data are bold. What are they stand for? Whether these two tables can be applied to data statistics.

Author Response

Reviewer 2

Catherine et al studied the relationship between BMI and workability at older ages. Data from 2299 men and 2425 women (aged 50-64 years) self-reported height and weight at baseline and provided information about workability at 12 and 24 months for the Health and Employment after Fifty (HEAF) Study. They found that obesity, and particularly severe obesity, negatively impacts workability at older ages with greatest effects in female workers. This cohort study is an interesting study. There are several issues the authors need to address.

We are grateful to the Reviewer for the positive feedback. 

The data is collected in 2013. Could the author find some recent data.

Our data are collected as part of a longitudinal cohort study, commencing in 2013-14 with annual follow-up thereafter.

In table 1 and 2, how to calculate the P value, should be announced in the manuscript. In table 3 and 4, some data are bold. What are they stand for? Whether these two tables can be applied to data statistics.

Thank you. The information has been inserted into the Tables. The bold denotes statistical significance at the p<0.05 level (95% level of confidence) and a note has been added to the Titles of the Tables accordingly.

Reviewer 3 Report

Authors should change the way they approach the problem, including several variables in structural models, and test the existing models and theories. The information here provided is very simple and not useful.

The problem of obesity is multifactorial, and its consequences equally multiple and correlated. I think that conducting studies on a single cause or consequence of obesity is currently completely irresponsible and biased since the variables overlap and hide between them, and more if the data were obtained by self-report. These types of studies should be carried out including many variables in multifactorial analysis and structural models, thereby seeing the influence and strength of each variable in the problem.
As observed in the results, the authors have other data that they should include in multifactorial analyzes and test the published models, reinforcing the published associations and/or providing new equations.
The association problems of only two variables are weak and subject to bias. Also, from a scientific point of view, partial publications, are biases and dishonest, creating many false positives and dangerous scientific myths.

Author Response

This prospective cohort study investigated the associations between body mass index (BMI) and workability. Over 4000 employed men and women aged 50-64 years were included in the data analysis. BMI was self-reported at baseline and participants were followed up by questionnaires for outcomes of interest including prolonged sickness absence, cutting down at work, as well as health-related job loss. The study found that obesity reduces workability after multiple adjustment at older ages especially in women. I found the study overall well designed and properly conducted. I only have a few minor comments/questions.

Thank you for your positive feedback.

Did the baseline questionnaire assess whether the participants had prolonged sickness absence or cutting down in the past (i.e., at baseline)?

Would you consider briefly explaining why the data analysis was conducted separately for men and women? This information might be interesting to readers.

Thank you for this helpful observation. Men and women do very different jobs throughout their careers and there is growing evidence that there are significant gender differences in many work outcomes. We have explained this in the text as follows:

Analyses were conducted separately for men and women because men and women undertake different types of occupations: e.g. men are more likely to be employed in construction /manufacture whilst women are more often employed in education, health and social care or retail.  

The sample size for severe obesity is relatively small especially for men.

We appreciate this comment and its possible influence on our findings and we have added a comment to this effect within the Discussion limitations:

It is important to note however, that we found a relatively low prevalence of severe obesity amongst the male respondents, which may have limited our statistical power to find associations in men that were seen in women.